# Path-Normalized Optimization of Recurrent Neural Networks with ReLU Activations

**Behnam Neyshabur**[*]
Toyota Technological Institute at Chicago
bneyshabur@ttic.edu

**Yuhuai Wu**[*]
University of Toronto
ywu@cs.toronto.edu

**Ruslan Salakhutdinov**
Carnegie Mellon University
rsalakhu@cs.cmu.edu

**Nathan Srebro**
Toyota Technological Institute at Chicago
nati@ttic.edu

## Abstract

We investigate the parameter-space geometry of recurrent neural networks (RNNs), and develop an adaptation of path-SGD optimization method, attuned to this geometry, that can learn plain RNNs with ReLU activations. On several datasets that require capturing long-term dependency structure, we show that path-SGD can significantly improve trainability of ReLU RNNs compared to RNNs trained with SGD, even with various recently suggested initialization schemes.

## 1 Introduction

Recurrent Neural Networks (RNNs) have been found to be successful in a variety of sequence learning problems [4, 3, 9], including those involving long term dependencies (e.g., [1, 23]). However, most of the empirical success has not been with "plain" RNNs but rather with alternate, more complex structures, such as Long Short-Term Memory (LSTM) networks [7] or Gated Recurrent Units (GRUs) [3]. Much of the motivation for these more complex models is not so much because of their modeling richness, but perhaps more because they seem to be easier to optimize. As we discuss in Section 3, training plain RNNs using gradient-descent variants seems problematic, and the choice of the activation function could cause a problem of vanishing gradients or of exploding gradients.

In this paper our goal is to better understand the geometry of plain RNNs, and develop better optimization methods, adapted to this geometry, that directly learn plain RNNs with ReLU activations. One motivation for insisting on plain RNNs, as opposed to LSTMs or GRUs, is because they are simpler and might be more appropriate for applications that require low-complexity design such as in mobile computing platforms [22, 5]. In other applications, it might be better to solve optimization issues by better optimization methods rather than reverting to more complex models. Better understanding optimization of plain RNNs can also assist us in designing, optimizing and intelligently using more complex RNN extensions.

Improving training RNNs with ReLU activations has been the subject of some recent attention, with most research focusing on different initialization strategies [12, 22]. While initialization can certainly have a strong effect on the success of the method, it generally can at most delay the problem of gradient explosion during optimization. In this paper we take a different approach that can be combined with any initialization choice, and focus on the dynamics of the optimization itself.

Any local search method is inherently tied to some notion of geometry over the search space (e.g. the space of RNNs). For example, gradient descent (including stochastic gradient descent) is tied to the Euclidean geometry and can be viewed as steepest descent with respect to the Euclidean norm. Changing the norm (even to a different quadratic norm, e.g. by representing the weights with respect to a different basis in parameter space) results in different optimization dynamics. We build on prior work on the geometry and optimization in feed-forward networks, which uses the path-norm [16]

---

[*]Contributed equally.

| | Input nodes | Internal nodes | Output nodes |
|---|---|---|---|
| FF (shared weights) | $h_v = x[v]$ | $h_v = \left[\sum_{(u \to v) \in E} w_{u \to v} h_u\right]_+$ | $h_v = \sum_{(u \to v) \in E} w_{u \to v} h_u$ |
| RNN notation | $\mathbf{h}_t^0 = \mathbf{x}_t, \mathbf{h}_0^i = 0$ | $\mathbf{h}_t^i = \left[\mathbf{W}_{\text{in}}^i h_t^{i-1} + \mathbf{W}_{\text{rec}}^i h_{t-1}^i\right]_+$ | $\mathbf{h}_t^d = \mathbf{W}_{\text{out}} \mathbf{h}_t^{d-1}$ |

Table 1: Forward computations for feedforward nets with shared weights.

(defined in Section 4) to determine a geometry leading to the path-SGD optimization method. To do so, we investigate the geometry of RNNs as feedforward networks with shared weights (Section 2) and extend a line of work on Path-Normalized optimization to include networks with shared weights. We show that the resulting algorithm (Section 4) has similar invariance properties on RNNs as those of standard path-SGD on feedforward networks, and can result in better optimization with less sensitivity to the scale of the weights.

## 2 Recurrent Neural Nets as Feedforward Nets with Shared Weights

We view Recurrent Neural Networks (RNNs) as feedforward networks with shared weights.

We denote a general feedforward network with ReLU activations and shared weights is indicated by $\mathcal{N}(G, \pi, \mathbf{p})$ where $G(V, E)$ is a directed acyclic graph over the set of nodes $V$ that corresponds to units $v \in V$ in the network, including special subsets of input and output nodes $V_{\text{in}}, V_{\text{out}} \subset V$, $\mathbf{p} \in \mathbb{R}^m$ is a parameter vector and $\pi : E \to \{1, \ldots, m\}$ is a mapping from edges in $G$ to parameters indices. For any edge $e \in E$, the weight of the edge $e$ is indicated by $w_e = p_{\pi(e)}$. We refer to the set of edges that share the $i$th parameter $p_i$ by $E_i = \{e \in E | \pi(e) = i\}$. That is, for any $e_1, e_2 \in E_i$, $\pi(e_1) = \pi(e_2)$ and hence $w_{e_1} = w_{e_2} = p_{\pi(e_1)}$.

Such a feedforward network represents a function $f_{\mathcal{N}(G, \pi, \mathbf{p})} : \mathbb{R}^{|V_{\text{in}}|} \to \mathbb{R}^{|V_{\text{out}}|}$ as follows: For any input node $v \in V_{\text{in}}$, its output $h_v$ is the corresponding coordinate of the input vector $\mathbf{x} \in \mathbb{R}^{|V_{\text{in}}|}$. For each internal node $v$, the output is defined recursively as $h_v = \left[\sum_{(u \to v) \in E} w_{u \to v} \cdot h_u\right]_+$ where $[z]_+ = \max(z, 0)$ is the ReLU activation function[2]. For output nodes $v \in V_{\text{out}}$, no non-linearity is applied and their output $h_v = \sum_{(u \to v) \in E} w_{u \to v} \cdot h_u$ determines the corresponding coordinate of the computed function $f_{\mathcal{N}(G, \pi, \mathbf{p})}(\mathbf{x})$. Since we will fix the graph $G$ and the mapping $\pi$ and learn the parameters $\mathbf{p}$, we use the shorthand $f_{\mathbf{p}} = f_{\mathcal{N}(G, \pi, \mathbf{p})}$ to refer to the function implemented by parameters $\mathbf{p}$. The goal of training is to find parameters $\mathbf{p}$ that minimize some error functional $L(f_{\mathbf{p}})$ that depends on $\mathbf{p}$ only through the function $f_{\mathbf{p}}$. E.g. in supervised learning $L(f) = \mathbb{E}\left[loss(f(x), y)\right]$ and this is typically done by minimizing an empirical estimate of this expectation.

If the mapping $\pi$ is a one-to-one mapping, then there is no weight sharing and it corresponds to standard feedforward networks. On the other hand, weight sharing exists if $\pi$ is a many-to-one mapping. Two well-known examples of feedforward networks with shared weights are *convolutional* and *recurrent* networks. We mostly use the general notation of feedforward networks with shared weights throughout the paper as this will be more general and simplifies the development and notation. However, when focusing on RNNs, it is helpful to discuss them using a more familiar notation which we briefly introduce next.

**Recurrent Neural Networks** Time-unfolded RNNs are feedforward networks with shared weights that map an input sequence to an output sequence. Each input node corresponds to either a coordinate of the input vector at a particular time step or a hidden unit at time 0. Each output node also corresponds to a coordinate of the output at a specific time step. Finally, each internal node refers to some hidden unit at time $t \geq 1$. When discussing RNNs, it is useful to refer to different layers and the values calculated at different time-steps. We use a notation for RNN structures in which the nodes are partitioned into layers and $\mathbf{h}_t^i$ denotes the output of nodes in layer $i$ at time step $t$. Let $\mathbf{x} = (\mathbf{x}_1, \ldots, \mathbf{x}_T)$ be the input at different time steps where $T$ is the maximum number of propagations through time and we refer to it as the length of the RNN. For $0 \leq i < d$, let $\mathbf{W}_{\text{in}}^i$ and $\mathbf{W}_{\text{rec}}^i$ be the input and recurrent parameter matrices of layer $i$ and $\mathbf{W}_{\text{out}}$ be the output parameter matrix. Table 1 shows forward computations for RNNs. The output of the function implemented by RNN can then be calculated as $f_{\mathbf{W}, t}(x) = h_t^d$. Note that in this notations, weight matrices $\mathbf{W}_{\text{in}}$, $\mathbf{W}_{\text{rec}}$ and $\mathbf{W}_{\text{out}}$ correspond to "free" parameters of the model that are shared in different time steps.

# 3    Non-Saturating Activation Functions

The choice of activation function for neural networks can have a large impact on optimization. We are particularly concerned with the distinction between "saturating" and "non-starting" activation functions. We consider only monotone activation functions and say that a function is "saturating" if it is bounded—this includes, e.g. sigmoid, hyperbolic tangent and the piecewise-linear ramp activation functions. Boundedness necessarily implies that the function values converge to finite values at negative and positive infinity, and hence asymptote to horizontal lines on both sides. That is, the derivative of the activation converges to zero as the input goes to both $-\infty$ and $+\infty$. Networks with saturating activations therefore have a major shortcoming: the vanishing gradient problem [6]. The problem here is that the gradient disappears when the magnitude of the input to an activation is large (whether the unit is very "active" or very "inactive") which makes the optimization very challenging.

While sigmoid and hyperbolic tangent have historically been popular choices for fully connected feedforward and convolutional neural networks, more recent works have shown undeniable advantages of non-saturating activations such as ReLU, which is now the standard choice for fully connected and Convolutional networks [15, 10]. Non-saturating activations, including the ReLU, are typically still bounded from below and asymptote to a horizontal line, with a vanishing derivative, at $-\infty$. But they are unbounded from above, enabling their derivative to remain bounded away from zero as the input goes to $+\infty$. Using ReLUs enables gradients to not vanish along activated paths and thus can provide a stronger signal for training.

However, for recurrent neural networks, using ReLU activations is challenging in a different way, as even a small change in the direction of the leading eigenvector of the recurrent weights could get amplified and potentially lead to the explosion in forward or backward propagation [1].

To understand this, consider a long path from an input in the first element of the sequence to an output of the last element, which passes through the same RNN edge at each step (i.e. through many edges in some $E_i$ in the shared-parameter representation). The length of this path, and the number of times it passes through edges associated with a single parameter, is proportional to the sequence length, which could easily be a few hundred or more. The effect of this parameter on the path is therefore exponentiated by the sequence length, as are gradient updates for this parameter, which could lead to parameter explosion unless an extremely small step size is used.

Understanding the geometry of RNNs with ReLUs could helps us deal with the above issues more effectively. We next investigate some properties of geometry of RNNs with ReLU activations.

**Invariances in Feedforward Nets with Shared Weights**

Feedforward networks (with or without shared weights) are highly over-parameterized, i.e. there are many parameter settings $\mathbf{p}$ that represent the same function $f_{\mathbf{p}}$. Since our true object of interest is the function $f$, and not the identity $\mathbf{p}$ of the parameters, it would be beneficial if optimization would depend only on $f_{\mathbf{p}}$ and not get "distracted" by difference in $\mathbf{p}$ that does not affect $f_{\mathbf{p}}$. It is therefore helpful to study the transformations on the parameters that will not change the function presented by the network and come up with methods that their performance is not affected by such transformations.

**Definition 1.** *We say a network $\mathcal{N}$ is* invariant *to a transformation $\mathcal{T}$ if for any parameter setting $\mathbf{p}$, $f_{\mathbf{p}} = f_{\mathcal{T}(\mathbf{p})}$. Similarly, we say an update rule $\mathcal{A}$ is* invariant *to $\mathcal{T}$ if for any $\mathbf{p}$, $f_{\mathcal{A}(\mathbf{p})} = f_{\mathcal{A}(\mathcal{T}(\mathbf{p}))}$.*

Invariances have also been studied as different mappings from the parameter space to the same function space [19] while we define the transformation as a mapping inside a fixed parameter space. A very important invariance in feedforward networks is *node-wise rescaling* [17]. For any internal node $v$ and any scalar $\alpha > 0$, we can multiply all incoming weights into $v$ (i.e. $w_{u \to v}$ for any $(u \to v) \in E$) by $\alpha$ and all the outgoing weights (i.e. $w_{v \to u}$ for any $(v \to u) \in E$) by $1/\alpha$ without changing the function computed by the network. Not all node-wise rescaling transformations can be applied in feedforward nets with shared weights. This is due to the fact that some weights are forced to be equal and therefore, we are only allowed to change them by the same scaling factor.

**Definition 2.** *Given a network $\mathcal{N}$, we say an invariant transformation $\widetilde{\mathcal{T}}$ that is defined over edge weights (rather than parameters) is* feasible *for parameter mapping $\pi$ if the shared weights remain equal after the transformation, i.e. for any $i$ and for any $e, e' \in E_i$, $\widetilde{\mathcal{T}}(\mathbf{w})_e = \widetilde{\mathcal{T}}(\mathbf{w})_{e'}$.*

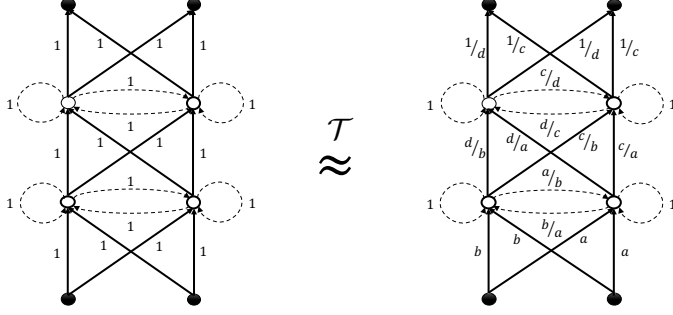

Figure 1: An example of invariances in an RNN with two hidden layers each of which has 2 hidden units. The dashed lines correspond to recurrent weights. The network on the left hand side is equivalent (i.e. represents the same function) to the network on the right for any nonzero $\alpha_1^1 = a$, $\alpha_2^1 = b$, $\alpha_1^2 = c$, $\alpha_2^2 = d$.

Therefore, it is helpful to understand what are the *feasible* node-wise rescalings for RNNs. In the following theorem, we characterize all feasible node-wise invariances in RNNs.

**Theorem 1.** *For any $\alpha$ such that $\alpha_j^i > 0$, any Recurrent Neural Network with ReLU activation is invariant to the transformation $\mathcal{T}_\alpha\left([\mathbf{W}_{in}, \mathbf{W}_{rec}, \mathbf{W}_{out}]\right) = \left[\mathcal{T}_{in,\alpha}\left(\mathbf{W}_{in}\right), \mathcal{T}_{rec,\alpha}\left(\mathbf{W}_{rec}\right), \mathcal{T}_{out,\alpha}\left(\mathbf{W}_{out}\right)\right]$ where for any $i, j, k$:*

$$\mathcal{T}_{in,\alpha}(\mathbf{W}_{in})^i[j,k] = \begin{cases} \alpha_j^i \mathbf{W}_{in}^i[j,k] & i = 1, \\ \left(\alpha_j^i / \alpha_k^{i-1}\right) \mathbf{W}_{in}^i[j,k] & 1 < i < d, \end{cases} \tag{1}$$

$$\mathcal{T}_{rec,\alpha}(\mathbf{W}_{rec})^i[j,k] = \left(\alpha_j^i / \alpha_k^i\right) \mathbf{W}_{rec}^i[j,k], \qquad \mathcal{T}_{out,\alpha}(\mathbf{W}_{out})[j,k] = \left(1/\alpha_k^{d-1}\right) \mathbf{W}_{out}[j,k].$$

*Furthermore, any feasible node-wise rescaling transformation can be presented in the above form.*

The proofs of all theorems and lemmas are given in the supplementary material. The above theorem shows that there are many transformations under which RNNs represent the same function. An example of such invariances is shown in Fig. 1. Therefore, we would like to have optimization algorithms that are invariant to these transformations and in order to do so, we need to look at measures that are invariant to such mappings.

## 4 Path-SGD for Networks with Shared Weights

As we discussed, optimization is inherently tied to a choice of geometry, here represented by a choice of complexity measure or "norm"[3]. Furthermore, we prefer using an invariant measure which could then lead to an invariant optimization method. In Section 4.1 we introduce the path-regularizer and in Section 4.2, the derived Path-SGD optimization algorithm for standard feed-forward networks. Then in Section 4.3 we extend these notions also to networks with shared weights, including RNNs, and present two invariant optimization algorithms based on it. In Section 4.4 we show how these can be implemented efficiently using forward and backward propagations.

### 4.1 Path-regularizer

The path-regularizer is the sum over all paths from input nodes to output nodes of the product of squared weights along the path. To define it formally, let $\mathcal{P}$ be the set of directed paths from input to output units so that for any path $\zeta = \left(\zeta_0, \ldots, \zeta_{\text{len}(\zeta)}\right) \in \mathcal{P}$ of length $\text{len}(\zeta)$, we have that $\zeta_0 \in V_{\text{in}}$, $\zeta_{\text{len}(\zeta)} \in V_{\text{out}}$ and for any $0 \leq i \leq \text{len}(\zeta) - 1$, $(\zeta_i \to \zeta_{i+1}) \in E$. We also abuse the notation and denote $e \in \zeta$ if for some $i$, $e = (\zeta_i, \zeta_{i+1})$. Then the path regularizer can be written as:

$$\gamma_{\text{net}}^2(\mathbf{w}) = \sum_{\zeta \in \mathcal{P}} \prod_{i=0}^{\text{len}(\zeta)-1} w_{\zeta_i \to \zeta_{i+1}}^2 \tag{2}$$

Equivalently, the path-regularizer can be defined recursively on the nodes of the network as:

$$\gamma_v^2(\mathbf{w}) = \sum_{(u \to v) \in E} \gamma_u^2(\mathbf{w}) w_{u \to v}^2, \qquad \gamma_{\text{net}}^2(\mathbf{w}) = \sum_{u \in V_{\text{out}}} \gamma_u^2(\mathbf{w}) \tag{3}$$

## 4.2 Path-SGD for Feedforward Networks

Path-SGD is an approximate steepest descent step with respect to the path-norm. More formally, for a network without shared weights, where the parameters are the weights themselves, consider the diagonal quadratic approximation of the path-regularizer about the current iterate $\mathbf{w}^{(t)}$:

$$\hat{\gamma}_{\text{net}}^2(\mathbf{w}^{(t)} + \Delta\mathbf{w}) = \gamma_{\text{net}}^2(\mathbf{w}^{(t)}) + \left\langle \nabla\gamma_{\text{net}}^2(\mathbf{w}^{(t)}), \Delta\mathbf{w} \right\rangle + \frac{1}{2}\Delta\mathbf{w}^\top \text{diag}\left( \nabla^2\gamma_{\text{net}}^2(\mathbf{w}^{(t)}) \right)\Delta\mathbf{w} \quad (4)$$

Using the corresponding quadratic norm $\|\mathbf{w} - \mathbf{w}'\|_{\hat{\gamma}_{\text{net}}^2(\mathbf{w}^{(t)}+\Delta\mathbf{w})}^2 = \frac{1}{2}\sum_{e\in E} \frac{\partial^2 \gamma_{\text{net}}^2}{\partial w_e^2}(w_e - w_e')^2$, we can define an approximate steepest descent step as:

$$\mathbf{w}^{(t+1)} = \min_{\mathbf{w}} \eta \left\langle \nabla L(\mathbf{w}), \mathbf{w} - \mathbf{w}^{(t)} \right\rangle + \left\| \mathbf{w} - \mathbf{w}^{(t)} \right\|_{\hat{\gamma}_{\text{net}}^2(\mathbf{w}^{(t)}+\Delta\mathbf{w})}^2. \quad (5)$$

Solving (5) yields the update:

$$w_e^{(t+1)} = w_e^{(t)} - \frac{\eta}{\kappa_e(\mathbf{w}^{(t)})}\frac{\partial L}{\partial w_e}(\mathbf{w}^{(t)}) \qquad \text{where: } \kappa_e(\mathbf{w}) = \frac{1}{2}\frac{\partial^2\gamma_{\text{net}}^2(\mathbf{w})}{\partial w_e^2}. \quad (6)$$

The stochastic version that uses a subset of training examples to estimate $\frac{\partial L}{\partial w_{u\to v}}(\mathbf{w}^{(t)})$ is called Path-SGD [16]. We now show how Path-SGD can be extended to networks with shared weights.

## 4.3 Extending to Networks with Shared Weights

When the networks has shared weights, the path-regularizer is a function of parameters $\mathbf{p}$ and therefore the quadratic approximation should also be with respect to the iterate $\mathbf{p}^{(t)}$ instead of $\mathbf{w}^{(t)}$ which results in the following update rule:

$$\mathbf{p}^{(t+1)} = \min_{\mathbf{p}} \eta \left\langle \nabla L(\mathbf{p}), \mathbf{p} - \mathbf{p}^{(t)} \right\rangle + \left\| \mathbf{p} - \mathbf{p}^{(t)} \right\|_{\hat{\gamma}_{\text{net}}^2(\mathbf{p}^{(t)}+\Delta\mathbf{p})}. \quad (7)$$

where $\|\mathbf{p} - \mathbf{p}'\|_{\hat{\gamma}_{\text{net}}^2(\mathbf{p}^{(t)}+\Delta\mathbf{p})}^2 = \frac{1}{2}\sum_{i=1}^m \frac{\partial^2\gamma_{\text{net}}^2}{\partial p_i^2}(p_i - p_i')^2$. Solving (7) gives the following update:

$$p_i^{(t+1)} = p_i^{(t)} - \frac{\eta}{\kappa_i(\mathbf{p}^{(t)})}\frac{\partial L}{\partial p_i}(\mathbf{p}^{(t)}) \qquad \text{where: } \kappa_i(\mathbf{p}) = \frac{1}{2}\frac{\partial^2\gamma_{\text{net}}^2(\mathbf{p})}{\partial p_i^2}. \quad (8)$$

The second derivative terms $\kappa_i$ are specified in terms of their path structure as follows:

**Lemma 1.** $\kappa_i(\mathbf{p}) = \kappa_i^{(1)}(\mathbf{p}) + \kappa_i^{(2)}(\mathbf{p})$ where

$$\kappa_i^{(1)}(\mathbf{p}) = \sum_{e\in E_i}\sum_{\zeta\in\mathcal{P}} \mathbf{1}_{e\in\zeta} \prod_{\substack{j=0 \\ e\neq(\zeta_j\to\zeta_{j+1})}}^{len(\zeta)-1} p_{\pi(\zeta_j\to\zeta_{j+1})}^2 = \sum_{e\in E_i}\kappa_e(\mathbf{w}), \quad (9)$$

$$\kappa_i^{(2)}(\mathbf{p}) = p_i^2 \sum_{\substack{e1,e2\in E_i \\ e_1\neq e_2}}\sum_{\zeta\in\mathcal{P}} \mathbf{1}_{e_1,e_2\in\zeta} \prod_{\substack{j=0 \\ e_1\neq(\zeta_j\to\zeta_{j+1}) \\ e_2\neq(\zeta_j\to\zeta_{j+1})}}^{len(\zeta)-1} p_{\pi(\zeta_j\to\zeta_{j+1})}^2, \quad (10)$$

and $\kappa_e(\mathbf{w})$ is defined in (6).

The second term $\kappa_i^{(2)}(\mathbf{p})$ measures the effect of interactions between edges corresponding to the same parameter (edges from the same $E_i$) on the same path from input to output. In particular, if for any path from an input unit to an output unit, no two edges along the path share the same parameter, then $\kappa^{(2)}(\mathbf{p}) = 0$. For example, for any feedforward or Convolutional neural network, $\kappa^{(2)}(\mathbf{p}) = 0$. But for RNNs, there certainly are multiple edges sharing a single parameter on the same path, and so we could have $\kappa^{(2)}(\mathbf{p}) \neq 0$.

The above lemma gives us a precise update rule for the approximate steepest descent with respect to the path-regularizer. The following theorem confirms that the steepest descent with respect to this regularizer is also invariant to all feasible node-wise rescaling for networks with shared weights.

**Theorem 2.** *For any feedforward networks with shared weights, the update* (8) *is invariant to all feasible node-wise rescalings. Moreover, a simpler update rule that only uses $\kappa_i^{(1)}(\mathbf{p})$ in place of $\kappa_i(\mathbf{p})$ is also invariant to all feasible node-wise rescalings.*

Equations (9) and (10) involve a sum over all paths in the network which is exponential in depth of the network. However, we next show that both of these equations can be calculated efficiently.

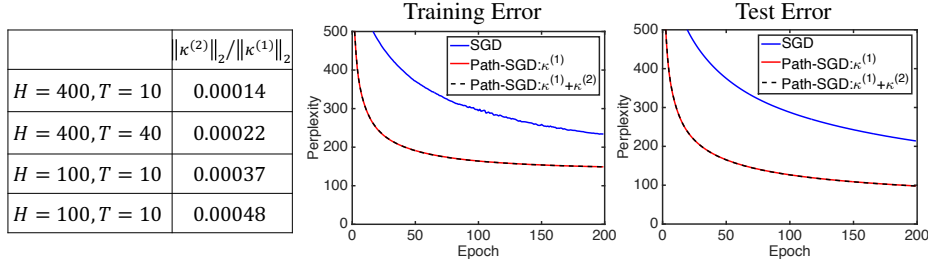

| | $\|\kappa^{(2)}\|_2/\|\kappa^{(1)}\|_2$ |
|---|---|
| $H = 400, T = 10$ | 0.00014 |
| $H = 400, T = 40$ | 0.00022 |
| $H = 100, T = 10$ | 0.00037 |
| $H = 100, T = 10$ | 0.00048 |

Figure 2: Path-SGD with/without the second term in word-level language modeling on PTB. We use the standard split (929k training, 73k validation and 82k test) and the vocabulary size of 10k words. We initialize the weights by sampling from the uniform distribution with range $[-0.1, 0.1]$. The table on the left shows the ratio of magnitude of first and second term for different lengths $T$ and number of hidden units $H$. The plots compare the training and test errors using a mini-batch of size 32 and backpropagating through $T = 20$ time steps and using a mini-batch of size 32 where the step-size is chosen by a grid search.

## 4.4 Simple and Efficient Computations for RNNs

We show how to calculate $\kappa_i^{(1)}(\mathbf{p})$ and $\kappa_i^{(2)}(\mathbf{p})$ by considering a network with the same architecture but with squared weights:

**Theorem 3.** *For any network $\mathcal{N}(G, \pi, p)$, consider $\mathcal{N}(G, \pi, \tilde{p})$ where for any $i$, $\tilde{p}_i = p_i^2$. Define the function $g : \mathbb{R}^{|V_{in}|} \to \mathbb{R}$ to be the sum of outputs of this network: $g(x) = \sum_{i=1}^{|V_{out}|} f_{\tilde{\mathbf{p}}}(x)[i]$. Then $\kappa^{(1)}$ and $\kappa^{(2)}$ can be calculated as follows where $\mathbf{1}$ is the all-ones input vector:*

$$\kappa^{(1)}(\mathbf{p}) = \nabla_{\tilde{\mathbf{p}}} g(\mathbf{1}), \qquad \kappa_i^{(2)}(\mathbf{p}) = \sum_{\substack{(u \to v),(u' \to v') \in E_i \\ (u \to v) \neq (u' \to v')}} \tilde{p}_i \frac{\partial g(\mathbf{1})}{\partial h_{v'}(\tilde{\mathbf{p}})} \frac{\partial h_{u'}(\tilde{\mathbf{p}})}{\partial h_v(\tilde{\mathbf{p}})} h_u(\tilde{\mathbf{p}}). \tag{11}$$

In the process of calculating the gradient $\nabla_{\tilde{\mathbf{p}}} g(\mathbf{1})$, we need to calculate $h_u(\tilde{\mathbf{p}})$ and $\partial g(\mathbf{1})/\partial h_v(\tilde{\mathbf{p}})$ for any $u, v$. Therefore, the only remaining term to calculate (besides $\nabla_{\tilde{p}} g(\mathbf{1})$) is $\partial h_{u'}(\tilde{\mathbf{p}})/\partial h_v(\tilde{\mathbf{p}})$.

Recall that $T$ is the length (maximum number of propagations through time) and $d$ is the number of layers in an RNN. Let $H$ be the number of hidden units in each layer and $B$ be the size of the mini-batch. Then calculating the gradient of the loss at all points in the minibatch (the standard work required for any mini-batch gradient approach) requires time $O(BdTH^2)$. In order to calculate $\kappa_i^{(1)}(\mathbf{p})$, we need to calculate the gradient $\nabla_{\tilde{\mathbf{p}}} g(1)$ of a similar network at a *single* input—so the time complexity is just an additional $O(dTH^2)$. The second term $\kappa^{(2)}(\mathbf{p})$ can also be calculated for RNNs in $O(dTH^2(T + H))$. For an RNN, $\kappa^{(2)}(\mathbf{W}_{\text{in}}) = 0$ and $\kappa^{(2)}(\mathbf{W}_{\text{out}}) = 0$ because only recurrent weights are shared multiple times along an input-output path. $\kappa^{(2)}(\mathbf{W}_{\text{rec}})$ can be written and calculated in the matrix form:

$$\kappa^{(2)}(\mathbf{W}_{\text{rec}}^i) = \mathbf{W}_{\text{rec}}'^i \odot \sum_{t_1=0}^{T-3} \left[ \left( (\mathbf{W}_{\text{rec}}'^i)^{t_1} \right)^\top \odot \sum_{t_2=2}^{T-t_1-1} \frac{\partial g(\mathbf{1})}{\partial \mathbf{h}_{t_1+t_2+1}^i(\tilde{\mathbf{p}})} \left( \mathbf{h}_{t_2}^i(\tilde{\mathbf{p}}) \right)^\top \right]$$

where for any $i, j, k$ we have $\mathbf{W}_{\text{rec}}'^i[j, k] = \left( \mathbf{W}_{\text{rec}}^i[j, k] \right)^2$. The only terms that require extra computation are powers of $\mathbf{W}_{\text{rec}}$ which can be done in $O(dTH^3)$ and the rest of the matrix computations need $O(dT^2H^2)$. Therefore, the ratio of time complexity of calculating the first term and second term with respect to the gradient over mini-batch is $O(1/B)$ and $O((T + H)/B)$ respectively. Calculating only $\kappa_i^{(1)}(\mathbf{p})$ is therefore very cheap with minimal per-minibatch cost, while calculating $\kappa_i^{(2)}(\mathbf{p})$ might be expensive for large networks. Beyond the low computational cost, calculating $\kappa_i^{(1)}(\mathbf{p})$ is also very easy to implement as it requires only taking the gradient with respect to a standard feed-forward calculation in a network with slightly modified weights—with most deep learning libraries it can be implemented very easily with only a few lines of code.

## 5 Experiments

### 5.1 The Contribution of the Second Term

As we discussed in section 4.4, the second term $\kappa^{(2)}$ in the update rule can be computationally expensive for large networks. In this section we investigate the significance of the second term

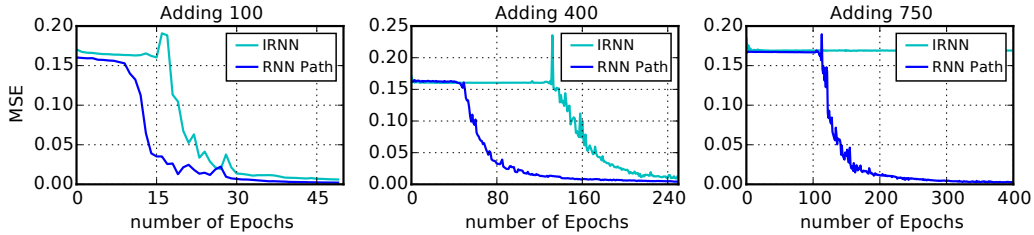

Figure 3: Test errors for the addition problem of different lengths.

and show that at least in our experiments, the contribution of the second term is negligible. To compare the two terms $\kappa^{(1)}$ and $\kappa^{(2)}$, we train a single layer RNN with $H = 200$ hidden units for the task of word-level language modeling on Penn Treebank (PTB) Corpus [13]. Fig. 2 compares the performance of SGD vs. Path-SGD with/without $\kappa^{(2)}$. We clearly see that both versions of Path-SGD are performing very similarly and both of them outperform SGD significantly. This results in Fig. 2 suggest that the first term is more significant and therefore we can ignore the second term.

To better understand the importance of the two terms, we compared the ratio of the norms $\left\|\kappa^{(2)}\right\|_2 / \left\|\kappa^{(1)}\right\|_2$ for different RNN lengths $T$ and number of hidden units $H$. The table in Fig. 2 shows that the contribution of the second term is bigger when the network has fewer number of hidden units and the length of the RNN is larger ($H$ is small and $T$ is large). However, in many cases, it appears that the first term has a much bigger contribution in the update step and hence the second term can be safely ignored. Therefore, in the rest of our experiments, we calculate the Path-SGD updates only using the first term $\kappa^{(1)}$.

## 5.2   Synthetic Problems with Long-term Dependencies

Training Recurrent Neural Networks is known to be hard for modeling long-term dependencies due to the gradient vanishing/exploding problem [6, 2]. In this section, we consider synthetic problems that are specifically designed to test the ability of a model to capture the long-term dependency structure. Specifically, we consider the addition problem and the sequential MNIST problem.

**Addition problem**: The addition problem was introduced in [7]. Here, each input consists of two sequences of length $T$, one of which includes numbers sampled from the uniform distribution with range $[0, 1]$ and the other sequence serves as a mask which is filled with zeros except for two entries. These two entries indicate which of the two numbers in the first sequence we need to add and the task is to output the result of this addition.

**Sequential MNIST**: In sequential MNIST, each digit image is reshaped into a sequence of length 784, turning the digit classification task into sequence classification with long-term dependencies [12, 1].

For both tasks, we closely follow the experimental protocol in [12]. We train a single-layer RNN consisting of 100 hidden units with path-SGD, referred to as **RNN-Path**. We also train an RNN of the same size with identity initialization, as was proposed in [12], using SGD as our baseline model, referred to as **IRNN**. We performed grid search for the learning rates over $\{10^{-2}, 10^{-3}, 10^{-4}\}$ for both our model and the baseline. Non-recurrent weights were initialized from the uniform distribution with range $[-0.01, 0.01]$. Similar to [1], we found the IRNN to be fairly unstable (with SGD optimization typically diverging). Therefore for IRNN, we ran 10 different initializations and picked the one that did not explode to show its performance.

In our first experiment, we evaluate Path-SGD on the addition problem. The results are shown in Fig. 3 with increasing the length $T$ of the sequence: $\{100, 400, 750\}$. We note that this problem becomes much harder as $T$ increases because the dependency between the output (the sum of two numbers) and the corresponding inputs becomes more distant. We also compare RNN-Path with the previously published results, including identity initialized RNN [12] (IRNN), unitary RNN [1] (uRNN), and np-RNN[4] introduced by [22]. Table 2 shows the effectiveness of using Path-SGD. Perhaps more surprisingly, with the help of path-normalization, a simple RNN with the identity initialization is able to achieve a 0% error on the sequences of length 750, whereas all the other methods, including LSTMs, fail. This shows that Path-SGD may help stabilize the training and alleviate the gradient problem, so as to perform well on longer sequence. We next tried to model

| | Adding 100 | Adding 400 | Adding 750 | sMNIST |
|---|---|---|---|---|
| IRNN [12] | 0 | 16.7 | 16.7 | 5.0 |
| uRNN [1] | 0 | 3 | 16.7 | 4.9 |
| LSTM [1] | 0 | 2 | 16.7 | 1.8 |
| np-RNN[22] | 0 | 2 | >2 | 3.1 |
| IRNN | 0 | 0 | 16.7 | 7.1 |
| RNN-Path | 0 | 0 | 0 | 3.1 |

Table 2: Test error (MSE) for the adding problem with different input sequence lengths and test classification error for the sequential MNIST.

| | PTB | text8 |
|---|---|---|
| RNN+smoothReLU [20] | - | 1.55 |
| HF-MRNN [14] | 1.42 | 1.54 |
| RNN-ReLU[11] | 1.65 | - |
| RNN-tanh[11] | 1.55 | - |
| TRec,$\beta = 500$[11] | 1.48 | - |
| RNN-ReLU | 1.55 | 1.65 |
| RNN-tanh | 1.58 | 1.70 |
| RNN-Path | 1.47 | 1.58 |
| LSTM | 1.41 | 1.52 |

Table 3: Test BPC for PTB and text8.

the sequences length of 1000, but we found that for such very long sequences RNNs, even with Path-SGD, fail to learn.

Next, we evaluate Path-SGD on the Sequential MNIST problem. Table 2, right column, reports test error rates achieved by RNN-Path compared to the previously published results. Clearly, using Path-SGD helps RNNs achieve better generalization. In many cases, RNN-Path outperforms other RNN methods (except for LSTMs), even for such a long-term dependency problem.

## 5.3 Language Modeling Tasks

In this section we evaluate Path-SGD on a language modeling task. We consider two datasets, Penn Treebank (PTB-c) and text8 [5]. **PTB-c**: We performed experiments on a tokenized Penn Treebank Corpus, following the experimental protocol of [11]. The training, validations and test data contain 5017k, 393k and 442k characters respectively. The alphabet size is 50, and each training sequence is of length 50. **text8**: The text8 dataset contains 100M characters from Wikipedia with an alphabet size of 27. We follow the data partition of [14], where each training sequence has a length of 180. Performance is evaluated using bits-per-character (BPC) metric, which is $\log_2$ of perplexity.

Similar to the experiments on the synthetic datasets, for both tasks, we train a single-layer RNN consisting of 2048 hidden units with path-SGD (RNN-Path). Due to the large dimension of hidden space, SGD can take a fairly long time to converge. Instead, we use Adam optimizer [8] to help speed up the training, where we simply use the path-SGD gradient as input to the Adam optimizer.

We also train three additional baseline models: a ReLU RNN with 2048 hidden units, a tanh RNN with 2048 hidden units, and an LSTM with 1024 hidden units, all trained using Adam. We performed grid search for learning rate over $\{10^{-3}, 5 \cdot 10^{-4}, 10^{-4}\}$ for all of our models. For ReLU RNNs, we initialize the recurrent matrices from uniform$[-0.01, 0.01]$, and uniform$[-0.2, 0.2]$ for non-recurrent weights. For LSTMs, we use orthogonal initialization [21] for the recurrent matrices and uniform$[-0.01, 0.01]$ for non-recurrent weights. The results are summarized in Table 3.

We also compare our results to an RNN that uses hidden activation regularizer [11] (TRec,$\beta = 500$), Multiplicative RNNs trained by Hessian Free methods [14] (HF-MRNN), and an RNN with smooth version of ReLU [20]. Table 3 shows that path-normalization is able to outperform RNN-ReLU and RNN-tanh, while at the same time shortening the performance gap between plain RNN and other more complicated models (e.g. LSTM by 57% on PTB and 54% on text8 datasets). This demonstrates the efficacy of path-normalized optimization for training RNNs with ReLU activation.

## 6 Conclusion

We investigated the geometry of RNNs in a broader class of feedforward networks with shared weights and showed how understanding the geometry can lead to significant improvements on different learning tasks. Designing an optimization algorithm with a geometry that is well-suited for RNNs, we closed over half of the performance gap between vanilla RNNs and LSTMs. This is particularly useful for applications in which we seek compressed models with fast prediction time that requires minimum storage; and also a step toward bridging the gap between LSTMs and RNNs.

**Acknowledgments**

This research was supported in part by NSF RI/AF grant 1302662, an Intel ICRI-CI award, ONR Grant N000141310721, and ADeLAIDE grant FA8750-16C-0130-001. We thank Saizheng Zhang for sharing a base code for RNNs.

## Footnotes

[2]The bias terms can be modeled by having an additional special node $v_{\text{bias}}$ that is connected to all internal and output nodes, where $h_{v_{\text{bias}}} = 1$.

[3]The path-norm which we define is a norm on functions, not on weights, but as we prefer not getting into this technical discussion here, we use the term "norm" very loosely to indicate some measure of magnitude [18].

[4]The original paper does not include any result for 750, so we implemented np-RNN for comparison. However, in our implementation the np-RNN is not able to even learn sequences of length of 200. Thus we put ">2" for length of 750.

[5]http://mattmahoney.net/dc/textdata

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
