[Supplementary Material]

# Supplementary: Path-Normalized Optimization of Recurrent Neural Networks with ReLU Activations

**Behnam Neyshabur**[*]
Toyota Technological Institute at Chicago
bneyshabur@ttic.edu

**Yuhuai Wu**[*]
University of Toronto
ywu@cs.toronto.edu

**Ruslan Salakhutdinov**
Carnegie Mellon University
rsalakhu@cs.cmu.edu

**Nathan Srebro**
Toyota Technological Institute at Chicago
nati@ttic.edu

## A  Proof of Theorem 1

We first show that any RNN is invariant to $\mathcal{T}_\alpha$ by induction on layers and time-steps. More specifically, we prove that for any $0 \leq t \leq T$ and $1 \leq i < d$, $\mathbf{h}_t^i\left(\mathcal{T}_\alpha(\mathbf{W})\right)[j] = \alpha_j^i \mathbf{h}_t^i(\mathbf{W})[j]$. The statement is clearly true for $t = 0$; because for any $i, j$, $\mathbf{h}_0^i\left(\mathcal{T}_\alpha(\mathbf{W})\right)[j] = \alpha_j^i \mathbf{h}_0^i(\mathbf{W})[j] = 0$.

Next, we show that for $i = 1$, if we assume that the statement is true for $t = t'$, then it is also true for $t = t' + 1$:

$$\mathbf{h}_{t'+1}^1\left(\mathcal{T}_\alpha(\mathbf{W})\right)[j] = \left[\sum_{j'} \mathcal{T}_{\text{in},\alpha}(\mathbf{W}_{\text{in}})^1[j,j']\mathbf{x}_{t'+1}[j'] + \mathcal{T}_{\text{rec},\alpha}(\mathbf{W}_{\text{rec}})^1[j,j']\mathbf{h}_{t'}^1\left(\mathcal{T}_\alpha(\mathbf{W})\right)[j']\right]_+$$

$$= \left[\sum_{j'} \alpha_j^1 \mathbf{W}_{\text{in}}^1[j,j']\mathbf{x}_{t'+1}[j'] + \left(\alpha_j^1/\alpha_{j'}^1\right)\mathbf{W}_{\text{rec}}^1[j,j']\alpha_{j'}^1\mathbf{h}_{t'}^1(\mathbf{W}))[j']\right]_+$$

$$= \alpha_j^1 \mathbf{h}_t^i(\mathbf{W})[j]$$

We now need to prove the statement for $1 < i < d$. Assuming that the statement is true for $t \leq t'$ and the layers before $i$, we have:

$$\mathbf{h}_{t'+1}^i\left(\mathcal{T}_\alpha(\mathbf{W})\right)[j] = \left[\sum_{j'} \mathcal{T}_{\text{in},\alpha}(\mathbf{W}_{\text{in}})^i[j,j']\mathbf{h}_{t'+1}^{i-1}\left(\mathcal{T}_\alpha(\mathbf{W})\right)[j'] + \mathcal{T}_{\text{rec},\alpha}(\mathbf{W}_{\text{rec}})^i[j,j']\mathbf{h}_{t'}^i\left(\mathcal{T}_\alpha(\mathbf{W})\right)[j']\right]_+$$

$$= \left[\sum_{j'} \frac{\alpha_j^i}{\alpha_{j'}^{i-1}}\mathbf{W}_{\text{in}}^i[j,j']\alpha_{j'}^{i-1}\mathbf{h}_{t'+1}^{i-1}(\mathbf{W}))[j'] + \frac{\alpha_j^i}{\alpha_{j'}^i}\mathbf{W}_{\text{rec}}^i[j,j']\alpha_{j'}^i\mathbf{h}_{t'}^i(\mathbf{W}))[j']\right]_+$$

$$= \alpha_j^i \mathbf{h}_t^i(\mathbf{W})[j]$$

Finally, we can show that the output is invariant for any $j$ at any time step $t$:

$$f_{\mathcal{T}(\mathbf{W}),t}(\mathbf{x}_t)[j] = \sum_{j'} \mathcal{T}_{\text{out},\alpha}(\mathbf{W}_{\text{out}})[j,j']\mathbf{h}_t^{d-1}(\mathcal{T}_\alpha(\mathbf{W})[j'] = \sum_{j'}(1/\alpha_{j'}^{d-1})\mathbf{W}_{\text{out}}[j,j']\alpha_{j'}^{d-1}\mathbf{h}_t^{d-1}(\mathbf{W})[j']$$

$$= \sum_{j'}\mathbf{W}_{\text{out}}[j,j']\mathbf{h}_t^{d-1}(\mathbf{W})[j'] = f_{\mathbf{W},t}(\mathbf{x}_t)[j]$$

---

[*]Contributed equally.

We now show that any feasible node-wise rescaling can be presented as $\mathcal{T}_\alpha$. Recall that node-wise rescaling invariances for a general feedforward network can be written as $\widetilde{\mathcal{T}_\beta}(\mathbf{w})_{u\to v} = (\beta_v/\beta_u)w_{u\to v}$ for some $\beta$ where $\beta_v > 0$ for internal nodes and $\beta_v = 1$ for any input/output nodes. An RNN with $T = 0$ has no weight sharing and for each node $v$ with index $j$ in layer $i$, we have $\beta_v = \alpha_j^i$. For any $T > 0$ however, we there is no invariance that is not already counted. The reason is that by fixing the values of $\beta_v$ for the nodes in time step $0$, due to the feasibility, the values of $\beta$ for nodes in other time-steps should be tied to the corresponding value in time step $0$. Therefore, all invariances are included and can be presented in form of $\mathcal{T}_\alpha$.

$\square$

## B   Proof of Lemma 1

We prove the statement simply by calculating the second derivative of the path-regularizer with respect to each parameter:

$$\kappa_i(\mathbf{p}) = \frac{1}{2}\frac{\partial^2 \gamma_{\mathrm{net}}^2}{\partial p_i^2} = \frac{1}{2}\frac{\partial}{\partial p_i}\left(\frac{\partial}{\partial p_i}\sum_{\zeta\in\mathcal{P}}\prod_{j=0}^{\mathrm{len}(\zeta)-1}w_{\zeta_j\to\zeta_{j+1}}^2\right)$$

$$= \frac{1}{2}\frac{\partial}{\partial p_i}\left(\frac{\partial}{\partial p_i}\sum_{\zeta\in\mathcal{P}}\prod_{j=0}^{\mathrm{len}(\zeta)-1}p_{\pi(\zeta_j\to\zeta_{j+1})}^2\right) = \frac{1}{2}\sum_{\zeta\in\mathcal{P}}\frac{\partial}{\partial p_i}\left(\frac{\partial}{\partial p_i}\prod_{j=0}^{\mathrm{len}(\zeta)-1}p_{\pi(\zeta_j\to\zeta_{j+1})}^2\right)$$

Taking the second derivative then gives us both terms after a few calculations:

$$\kappa_i(\mathbf{p}) = \frac{1}{2}\sum_{\zeta\in\mathcal{P}}\frac{\partial}{\partial p_i}\left(\frac{\partial}{\partial p_i}\prod_{j=0}^{\mathrm{len}(\zeta)-1}p_{\pi(\zeta_j\to\zeta_{j+1})}^2\right) = \sum_{\zeta\in\mathcal{P}}\frac{\partial}{\partial p_i}\left(p_i\sum_{e\in E_i}\mathbf{1}_{e\in\zeta}\prod_{\substack{j=0\\e\neq(\zeta_j\to\zeta_{j+1})}}^{\mathrm{len}(\zeta)-1}p_{\pi(\zeta_j\to\zeta_{j+1})}^2\right)$$

$$= \sum_{\zeta\in\mathcal{P}}\left[p_i\frac{\partial}{\partial p_i}\left(\sum_{e\in E_i}\mathbf{1}_{e\in\zeta}\prod_{\substack{j=0\\e\neq(\zeta_j\to\zeta_{j+1})}}^{\mathrm{len}(\zeta)-1}p_{\pi(\zeta_j\to\zeta_{j+1})}^2\right) + \sum_{e\in E_i}\mathbf{1}_{e\in\zeta}\prod_{\substack{j=0\\e\neq(\zeta_j\to\zeta_{j+1})}}^{\mathrm{len}(\zeta)-1}p_{\pi(\zeta_j\to\zeta_{j+1})}^2\right]$$

$$= p_i^2\sum_{\substack{e1,e2\in E_i\\e_1\neq e_2}}\left[\sum_{\zeta\in\mathcal{P}}\mathbf{1}_{e_1,e_2\in\zeta}\prod_{\substack{j=0\\e_1\neq(\zeta_j\to\zeta_{j+1})\\e_2\neq(\zeta_j\to\zeta_{j+1})}}^{\mathrm{len}(\zeta)-1}p_{\pi(\zeta_j\to\zeta_{j+1})}^2\right] + \sum_{e\in E_i}\left[\sum_{\zeta\in\mathcal{P}}\mathbf{1}_{e\in\zeta}\prod_{\substack{j=0\\e\neq(\zeta_j\to\zeta_{j+1})}}^{\mathrm{len}(\zeta)-1}p_{\pi(\zeta_j\to\zeta_{j+1})}^2\right]$$

$\square$

## C   Proof of Theorem 2

Node-wise rescaling invariances for a feedforward network can be written as $\mathcal{T}_\beta(\mathbf{w})_{u\to v} = (\beta_v/\beta_u)w_{u\to v}$ for some $\beta$ where $\beta_v > 0$ for internal nodes and $\beta_v = 1$ for any input/output nodes. Any feasible invariance for a network with shared weights can also be written in the same form. The only difference is that some of $\beta_v$s are now tied to each other in a way that shared weights have the same value after transformation. First, note that since the network is invariant to the transformation, the following statement holds by an induction similar to Theorem 1 but in the backward direction:

$$\frac{\partial L}{\partial h_v}(\mathcal{T}_\beta(\mathbf{p})) = \frac{1}{\beta_v}\frac{\partial L}{\partial h_u}(\mathbf{p}) \tag{1}$$

for any $(u \to v) \in E$. Furthermore, by the proof of the Theorem 1 we have that for any $(u \to v) \in E$, $h_u(\mathcal{T}_\beta(\mathbf{p})) = \beta_u h_u(\mathbf{p})$. Therefore,

$$\frac{\partial L}{\partial \mathcal{T}_\beta(\mathbf{p})_i}(\mathcal{T}_\beta(\mathbf{p})) = \sum_{(u\to v)\in E_i}\frac{\partial L}{\partial h_v}(\mathcal{T}_\beta(\mathbf{p}))h_u(\mathcal{T}_\beta(\mathbf{p})) = \frac{\beta_{u'}}{\beta_{v'}}\frac{\partial L}{\partial p_i}(\mathbf{p}) \tag{2}$$

where $(u' \rightarrow v') \in E_i$. In order to prove the theorem statement, it is enough to show that for any edge $(u \rightarrow v) \in E_i$, $\kappa_i(\mathcal{T}_\beta(\mathbf{p})) = (\beta_u/\beta_v)^2 \kappa_i(\mathbf{p})$ because this property gives us the following update:

$$\mathcal{T}_\beta(\mathbf{p})_i - \frac{\eta}{\kappa_i(\mathcal{T}_\beta(\mathbf{p}))} \frac{\partial L(\mathcal{T}_\beta(\mathbf{p}))}{\partial \mathcal{T}_\beta(\mathbf{p})_i} = \frac{\beta_v}{\beta_u} p_i - \frac{\eta}{(\beta_u/\beta_v)^2 \kappa_i(\mathbf{p})} \frac{\beta_u}{\beta_v} \frac{\partial L}{\partial p_i}(\mathbf{p}) = \mathcal{T}_\beta(\mathbf{p}^+)_i$$

Therefore, it is remained to show that for any edge $(u \rightarrow v) \in E_i$ $v$, $\kappa_i(\mathcal{T}_\beta(\mathbf{p})) = (\beta_u/\beta_v)^2 \kappa_i(\mathbf{p})$. We show that this is indeed true for both terms $\kappa^{(1)}$ and $\kappa^{(2)}$ separately.

We first prove the statement for $\kappa^{(1)}$. Consider each path $\zeta \in \mathcal{P}$. By an inductive argument along the path, it is easy to see that multiplying squared weights along this path is invariant to the transformation:

$$\prod_{j=0}^{\text{len}(\zeta)-1} \mathcal{T}_\beta(\mathbf{p})^2_{\pi(\zeta_j \rightarrow \zeta_{j+1})} = \prod_{j=0}^{\text{len}(\zeta)-1} p^2_{\pi(\zeta_j \rightarrow \zeta_{j+1})}$$

Therefore, we have that for any edge $e \in E$ and any $\zeta \in \mathcal{P}$,

$$\prod_{\substack{j=0 \\ e \neq (\zeta_j \rightarrow \zeta_{j+1})}}^{\text{len}(\zeta)-1} \mathcal{T}_\beta(\mathbf{p})^2_{\pi(\zeta_j \rightarrow \zeta_{j+1})} = \left(\frac{\beta_u}{\beta_v}\right)^2 \prod_{\substack{j=0 \\ e \neq (\zeta_j \rightarrow \zeta_{j+1})}}^{\text{len}(\zeta)-1} p^2_{\pi(\zeta_j \rightarrow \zeta_{j+1})}$$

Taking sum over all paths $\zeta \in \mathcal{P}$ and all edges $e = (u \rightarrow v) \in E$ completes the proof for $\kappa^{(1)}$. Similarly for $\kappa^{(2)}$, considering any two edges $e_1 \neq e_2$ and any path $\zeta_\mathcal{P}$, we have that:

$$\mathcal{T}_\beta(\mathbf{p})^2_i \prod_{\substack{j=0 \\ e_1 \neq (\zeta_j \rightarrow \zeta_{j+1}) \\ e_2 \neq (\zeta_j \rightarrow \zeta_{j+1})}}^{\text{len}(\zeta)-1} \mathcal{T}_\beta(\mathbf{p})^2_{\pi(\zeta_j \rightarrow \zeta_{j+1})} = \left(\frac{\beta_v}{\beta_u}\right)^2 p^2_i \left(\frac{\beta_u}{\beta_v}\right)^4 \prod_{\substack{j=0 \\ e_1 \neq (\zeta_j \rightarrow \zeta_{j+1}) \\ e_2 \neq (\zeta_j \rightarrow \zeta_{j+1})}}^{\text{len}(\zeta)-1} p^2_{\pi(\zeta_j \rightarrow \zeta_{j+1})}$$

where $(u \rightarrow v) \in E_i$. Again, taking sum over all paths $\zeta$ and all edges $e_1 \neq e_2$ proves the statement for $\kappa^{(2)}$ and consequently for $\kappa^{(1)} + \kappa^{(2)}$.

$\square$

# D  Proof of Theorem 3

First, note that based on the definitions in the theorem statement, for any node $v$, $h_v(\tilde{\mathbf{p}}) = \gamma_v^2(p)$ and therefore $g(\mathbf{1}) = \gamma_{\text{net}}^2(p)$. Using Lemma 1, main observation here is that for each edge $e \in E_i$ and each path $\zeta \in \mathcal{P}$, the corresponding term in $\kappa^{(1)}$ is nothing but product of the squared weights along the path except the weights that correspond to the edge $e$:

$$\mathbf{1}_{e \in \zeta} \prod_{\substack{j=0 \\ e \neq (\zeta_j \rightarrow \zeta_{j+1})}}^{\text{len}(\zeta)-1} p^2_{\pi(\zeta_j \rightarrow \zeta_{j+1})}$$

This path can therefore be decomposed into a path from input to edge $e$ and a path from edge $e$ to the output. Therefore, for any edge $e$, we can factor out the number corresponding to the paths that go through $e$ and rewrite $\kappa^{(1)}$ as follows:

$$\kappa^{(1)}(p) = \sum_{(u \rightarrow v) \in E_i} \left[ \left( \sum_{\zeta \in \mathcal{P}_{\text{in} \rightarrow u}} \prod_{j=0}^{\text{len}(\zeta)-1} p^2_{\pi(\zeta_j \rightarrow \zeta_{j+1})} \right) \left( \sum_{\zeta \in \mathcal{P}_{v \rightarrow \text{out}}} \prod_{j=0}^{\text{len}(\zeta)-1} p^2_{\pi(\zeta_j \rightarrow \zeta_{j+1})} \right) \right] \quad (3)$$

where $\mathcal{P}_{\text{in} \rightarrow u}$ is the set of paths from input nodes to node $v$ and $\mathcal{P}_{v \rightarrow \text{out}}$ is defined similarly for the output nodes.

By induction on layers of $\mathcal{N}(G, \pi, \tilde{\mathbf{p}})$, we get the following:

$$\sum_{\zeta \in \mathcal{P}_{\text{in} \rightarrow u}} \prod_{j=0}^{\text{len}(\zeta)-1} p^2_{\pi(\zeta_j \rightarrow \zeta_{j+1})} = h_u(\tilde{\mathbf{p}}) \quad (4)$$

$$\sum_{\zeta \in \mathcal{P}_{v \rightarrow \text{out}}} \prod_{j=0}^{\text{len}(\zeta)-1} p^2_{\pi(\zeta_j \rightarrow \zeta_{j+1})} = \frac{\partial g(\mathbf{1})}{\partial h_v(\tilde{\mathbf{p}})} \quad (5)$$

Therefore, $\kappa^{(1)}$ can be written as:

$$\kappa^{(1)}(p) = \sum_{(u \to v) \in E_i} \frac{\partial g(1)}{\partial h_v(\tilde{\mathbf{p}})} h_u(\tilde{\mathbf{p}}) = \sum_{(u \to v) \in E_i} \frac{\partial g(1)}{\partial w'_{u \to v}} = \frac{\partial g(1)}{\partial \tilde{p}_i} \tag{6}$$

Next, we show how to calculate the second term, i.e. $\kappa^{(2)}$. Each term in $\kappa^{(2)}$ corresponds to a path that goes through two edges. We can decompose such paths and rewrite $\kappa^{(2)}$ similar to the first term:

$$\kappa^{(2)}(p) = p_i^2 \sum_{\substack{(u \to v) \in E_i \\ (u' \to v') \in E_i \\ (u \to v) \neq (u' \to v')}} \left[ \left( \sum_{\zeta \in \mathcal{P}_{\text{in} \to u}} \prod_{j=0}^{\text{len}(\zeta)} p_{\pi(\zeta_j \to \zeta_{j+1})}^2 \right) \right.$$

$$\left( \sum_{\zeta \in \mathcal{P}_{v \to u'}} \prod_{j=0}^{\text{len}(\zeta)-1} p_{\pi(\zeta_j \to \zeta_{j+1})}^2 \right) \left( \sum_{\zeta \in \mathcal{P}_{v' \to \text{out}}} \prod_{j=0}^{\text{len}(\zeta)-1} p_{\pi(\zeta_j \to \zeta_{j+1})}^2 \right) \right]$$

$$= \sum_{\substack{(u \to v) \in E_i \\ (u' \to v') \in E_i \\ (u \to v) \neq (u' \to v')}} \tilde{p}_i \frac{\partial g(\mathbf{1})}{\partial h_{v'}(\tilde{\mathbf{p}})} \frac{\partial h_{u'}(\tilde{\mathbf{p}})}{\partial h_v(\tilde{\mathbf{p}})} h_u(\tilde{\mathbf{p}})$$

where $\mathcal{P}_{u \to v}$ is the set of all directed paths from node $u$ to node $v$.

$\square$