[Reviews · NeurIPS 2016]

Reviewer 1

Summary

The idea of path-SGD is applied to RNNs. Path SGD differs from more traditional regularisers (such as L2 weight decay), by taking a particular form which is a sum over paths from input to output nodes (see equation 2). It is shown that the regulariser remains invariant to those rescalings of the parameters which leave the function unchanged. An approximation from a previous paper is adapted to deal with recurrence as in RNNs. Furthermore, it is shown that the regulariser is dominated (in practice) by the cheaper of two terms which need computing, hence this further approximation is recommended and justified empirically. Experimental results are provided which, while they don’t obtain the level of LSTM etc, provide a significant improvement for vanilla RNN, with it’s associated advantages of simplicity etc.

Qualitative Assessment

I enjoyed reading this paper. Employing a norm with the demonstrated invariances, is extremely interesting and has the potential for big impact in the neural net community. This is one of several papers pushing this idea, and the only argument might be that the advance here is perhaps overly incremental. Nonetheless, that we can apply this idea to RNN, and that it is so effective here, is good to know.

Confidence in this Review

2-Confident (read it all; understood it all reasonably well)


Reviewer 2

Summary

This paper addresses issues with training RNNs with non-saturating activations functions (e.g., RELU). They present a theoretical justification by adopting the a general framework of a feed-forward network with shared parameters (an RNN unrolled infinitely has a single parameter for all steps). They present a path-SGD algorithm which allows one to train these models w/o the vanishing or exploding graident problem. They show empirically that while there is an additional term to compute for the path-norma gradient update, it does not impact training time. They also show that the path-norm sgd training is far superior to standard SGD training.

Qualitative Assessment

The paper is well written but the notation is tough to follow and the lack of diagrams makes the geometric nature of the problem more difficult to understand than necessary. I think that a few sketches of what is going on would help in interpreting the path algorithm. As I understand it, the path-norm is used to control the magnitude of the gradients. It also appears to be a good way to speed up learning as it renormalizes the gradients during learning.

Confidence in this Review

1-Less confident (might not have understood significant parts)


Reviewer 3

Summary

This is an interesting paper that attempts to understand the geometry of plain RNNs and proposes a path-SGD optimization method for plain RNNs with ReLU activations. The authors start by viewing RNNs as feedforward networks with shared weights, and then explore the invariances in feedforward networks with shared weights, especially for RNNs with ReLU activations. The authors further propose a path-regularizer and path-SGD optimization for feedforward networks with shared weights (RNNs). Experiments on both synthetic problems and real language modeling tasks illustrate the effectiveness of the proposed path-SGD method. Generally, I think the paper does a good job of advocating the importance of investigating the geometry of plain RNNs and designing optimization algorithm with a geometry that suits the RNNs. The paper, however, does not convince me that similar strategies will improve more advanced RNNs such as LSTMs.

Qualitative Assessment

1. This may not be a fair comment to the authors, but the first impression when I read the paper was like: why RNNs? why not LSTMs? I understand that LSTMs are more resource-intensive, but they are the main stream in a lot of applications due to their superior performance. So why don't the authors investigate the geometry of LSTMs instead of plain RNNs? Sorry, but this was my first impression... 2. Since the paper is mostly focusing on plain RNNs with ReLU activations, I think it might make sense to conduct experiments on more practical tasks, e.g., speech recognition tasks (toolkits such as Kaldi should be very helpful). This way the authors can really convince the readers of the effectiveness of the proposed path-SGD to plain RNNs with ReLU. 3. It might make sense to go beyond plain RNNs with ReLU. As a theory paper, I think the focus on plain RNN with ReLU activation is too limited. But as a practical paper, I'd like to see more experiments. 4. Good results on both synthetic problems and the real language modeling task!

Confidence in this Review

1-Less confident (might not have understood significant parts)


Reviewer 4

Summary

In this paper, the authors attempt to extend upon a previous work on path-normalization for regular feedforward neural nets into the regular (plain) RNNs using ReLUs. In the formulation, they cast RNNs as a fully unfolded feedforward neural nets with shared weights. The motivation is straightforward. However, the formulation only works for RNNs using ReLU (not for sigmoid RNNs and so on). It is well known that ReLUs work extremely well for feedforward NNs, such as DNNs and CNNs. Some work was reported for ReLU RNNs as well, however, we have NEVER been able to make ReLU RNNs work as well as regular sigmoid (or tanh) RNNs, not to mention LSTMs. The theoretical formulation to extend path-norm to RNNs seems straightforward. However, it is still critical to expose both strength and weakness of your proposed approaches to readers. I have two concerns: (1) Regarding the complexity of your path norm: you mention that the motivation to study plain RNNs rather than LSTMs is due to the simplicity. However, when you apply the path-norm to optimize RNNs, it seems you have to compute both k1 and k2 for every single data sample. This seems prohibitedly expensive, comparing with a simple SGD. (2) Performance your path norm: in section 5.3, you study your method for two popular language modelling, PTB and text8. Oddly enough, you report bits-per-character (BPC) for these tasks, rather than the standard word-level perplexity values. These two tasks have been widely studied and the state-of-the-art performance is well known based on the word level perplexity. I don't know why you choose a non-standard metric for these. If you have nothing to hide, please report the results in terms of word PPL. In this way, readers may easily compare your models and methods with tons of work in the literature. I INSIST this as a pre-requisite to accept this paper for publication.

Qualitative Assessment

As per my comments above: 1. Address the computational complexity issues of your path-norm. 2. Report your LM performance using the standard PPL.

Confidence in this Review

3-Expert (read the paper in detail, know the area, quite certain of my opinion)


Reviewer 5

Summary

In this paper, the path-SGD optimization method for feed-forward networks is extended to ReLU RNNs. Experimental results on sequential MNIST and word-level language modeling on PTB corpus show that the RNNs trained with the proposed path-SGD outperform IRNNs.

Qualitative Assessment

The authors extended the path-SGD to ReLU RNNs. Although some parts (e.g. proofs) of the paper are beyond my understanding, the flow of the paper is very logical and the experimental results are convincing. As shown in Section 4.4, the proposed method is computationally cheap when the second term (k^(2)) is ignored, and authors showed by experiments that the second term is not needed in practice.

Confidence in this Review

1-Less confident (might not have understood significant parts)


Reviewer 6

Summary

In this work the authors present an adaptation of the Path-SGD algorithm for RNNs. They characterize all node-wise invariances in RNNs and show that the update rule for their modified algorithm is invariant to all feasible node-wise rescalings. They provide experiments which indicate that their modified optimization method yields improvements for plain RNNs trained with existing methods, although they still lag behind LSTMs.

Qualitative Assessment

The authors frame their work as a step towards bridging the gap between plain RNNs and more complex LSTMs/GRUs by using an optimization method which accounts for the geometry of the loss surface. This seems to be a worthwhile goal (since plain RNNs are computationally cheaper and easier to analyze theoretically) and their experiments show some promising results in improving performance over plain RNNs trained with existing optimization methods. However, it is not clear to me how the method that the authors use in practice differs significantly from regular Path-SGD introduced in previous work. The authors do present an adaptation of Path-SGD to networks with shared weights, and show that the new rescaling term applied to the gradients can be divided into two terms k1 and k2. But then, they note that the second term, which accounts for interactions between shared weights along the same path, is expensive to calculate for RNNs and show some empirical evidence that including it does not help performance. In the rest of the experiments, they ignore the second term, which to my understanding is essentially what makes the method introduced here different from regular Path-SGD. Isn't ignoring the second term the same as simply applying Path-SGD as if the weights were not shared, and combining the rescaling terms corresponding to shared weights? Also, part of the justification for this work in Section 3 is that RNNs with ReLUs can encounter the "exploding activation" problem due to the fact that ReLUs do not saturate, and that taking into account the interactions between shared weights along the same path can avoid this problem. I would have thought the second term would be necessary to avoid this. I would like the authors to clarify the major differences, if any, between the approximation that they use in the experiments (which ignores the second term) and the Path-SGD method introduced in other work, and its relation to the exploding activation problem in ReLU-RNNs. I would be willing to raise my score if the authors can explain how the method that they use in practice is significantly different than the previous Path-SGD algorithm. The experiments do indicate that using path normalization offers a consistent improvement for plain RNNs on language modeling and sequential MNIST, which is encouraging, although they have yet to catch up to the LSTM. A few remarks: -Section 5.1: it would be better to report experiments on the same tasks that are used in evaluation, i.e. character level rather than word-level experiments. - for addition problem experiments, it seems like the authors initialize the RNN with identity transition matrix (this could be made more clear). Note that this is close to the exact solution presented in "Recurrent Orthogonal Networks and Long Memory Tasks", Henaff et al. 2016, and gives the RNN an advantage. Also, the experiments in that paper show that LSTMs can solve the addition task for T=750. For fair comparison with LSTMs, the RNN should be initialized randomly. -Line 217: both version -> both versions

Confidence in this Review

1-Less confident (might not have understood significant parts)